# Phylogeography of the Plateau Pika (*Ochotona curzoniae*) in Response to the Uplift of the Qinghai-Tibet Plateau

Yinglian Qi [1], Xiaoyan Pu [2], Zhilian Li [3], Daoguang Song [1] and Zhi Chen [1,3,*]

1 School of Geographic Science, Qinghai Normal University, Xining 810008, China
2 Medical College, Qinghai University, Xining 810008, China
3 School of Life Science, Qinghai Normal University, Xining 810008, China
* Correspondence: czi58@163.com; Tel.: +86-13099780863

**Abstract:** The evolution and current distribution of species on the Qinghai-Tibet Plateau have been significantly impacted by historical occurrences, including the uplift of the plateau and the Quaternary climate upheaval. As a remnant species, the plateau pika (*Ochotona curzoniae*) is a great model for researching historical events. In this study, 302 samples from 42 sample sites were utilized to analyze the impact of historical events on the evolution and distribution pattern of plateau pikas. The genetic diversity, patterns of differentiation, and historical dynamics of the plateau pika were investigated using molecular markers that included four mitochondrial genes (*COI*, *D-loop*, *Cytb*, and *12S rRNA*) and three nuclear genes (*GHR*, *IRBP*, and *RAG1*). The results showed that: (1) The genetic diversity of the plateau pika was high in the Tibetan Plateau ($H_d = 0.9997$, $\pi = 0.01205$), and the plateau pika evolved into five lineages that occupied different geographical areas, with lineage 1 (Group 1) in the south of the Yarlung Zangbo River, lineage 2 (Group 2) in the hinterland of the plateau, lineage 3 (Group 3) in the northeastern part of the plateau, lineage 4 (Group 4) in the Hengduan Mountains, and lineage 5 (Group 5) in the eastern part of the plateau. (2) The gene flow among the five lineages was low, and the differentiation level was high ($N_m < 0.25$; $Fst > 0.25$), indicating that the geographical barriers between the five lineages, such as the Yarlung Zangbo River, the Qaidam-Ghuong-Guide Basin, and the Lancang River, effectively promoted the population differentiation of the plateau pika. (3) The plateau pika first spread from the Hengduan Mountains to the entire Qinghai-Tibet Plateau and then conducted small-scale migration and dispersal in several refuges across the plateau in response to climate changes during the glacial and interglacial periods. (4) Except for Group 1 and Group 4, all the other populations exhibited a rapid expansion between 0.06 and 0.01 Mya, but the expansion was considerably delayed or halted by the effects of climate change during the last glacial maximum (0.02 Mya). Overall, the plateau pika on the Qinghai-Tibet Plateau exhibits high genetic diversity, and topographic obstacles, including mountains, valleys, and basins, created by the uplift of the plateau and climatic changes since the Quaternary period have played an important role in the differentiation and historical dynamics of the plateau pika population.

**Keywords:** plateau pika; phylogenetic; history dynamic; geographic barrier; refuge





## 1. Introduction

The Qinghai-Tibet Plateau, which spans an area of over 2.5 million km² and has an average elevation of 4000 m, is situated in Southwest China [1]. The Himalaya Mountains, Gangdise Mountains, Nianqing Tanggula Mountains, Tanggula Mountains, and Kunlun Mountains are the principal mountain ranges that run from south to north. The Yellow River, Yarlung Zangbo River, Jinsha River, and Salween River are the principal rivers. Global climate change and Cenozoic geological occurrences resulted in the area's high elevation and complicated terrain. According to Shi et al. [2], the Qinghai-Tibet Plateau experienced two uplifting and leveling cycles in the Tertiary period. The Gangdisi Mountain

uplift resulted from the first uplifting, which took place between 45–38 Mya, and the Himalayan Mountain uplift resulted from the second uplifting, which took place 30–21.8 Mya. Following the uplift, the Tibetan Plateau's average elevation was about 1500–2000 m during this time, but after a protracted period of denudation and leveling, the elevation of the Qinghai-Tibet plateau was only around 1000 m, although at this time it was still a tropical and subtropical environment [3]. The Qinghai-Tibet Plateau experienced a third dramatic uplift and complex climate changes during the late Mesozoic and Quaternary periods. The changes in this period were the main causes of its current complex topography. The third uplift has been divided into three stages by geologists. The Qinghai-Tibet Plateau rose to a height of 3000–3500 m during the first stage of the Qinghai-Tibet Movement (3.4–1.3 Mya) and the second stage of the Kung-Huang Movement (1.1–0.6 Mya). During the third stage, it rose to a height of 4000 m, which is known as the Republic Movement (0.15 Mya to today). During the dramatic uplift of the plateau in the Quaternary period, the plateau experienced several glaciations, which resulted in the gradual succession of the species-rich tropical forest to sparse grassland or even bare land. In addition, the circulation of the glacial and interglacial periods led to active water erosion in the plateau area during this period, and the flows of the Yarlung Zangbo River, Yangtze River, Yellow River, and other rivers were cut down sharply, forming many gorges [2]. These high mountains and deep gorges formed obvious geographical barriers and significantly affected the inheritance and differentiation of the relict species in the plateau areas [4–6]. Yu and Zhang [7] summarized the results of phylogeographical studies conducted on 40 plant species and found that most of the plants that are distributed in the northeastern part of the Qinghai-Tibet Plateau experienced an expansion to the plateau hinterland after the last glacial age. The main clades of most of the species that are distributed in the southeastern part of the Qinghai-Tibet Plateau diverged in the early Quaternary due to the uplift of the plateau, and they survived in several refuges on the edge of the plateau or expanded into small areas in response to climate change during the glacial period. Some plants also showed completely different responses to climate change during the ice age. For example, a few hardy plants, such as *Potentilla glabra* [8], did not retreat but expanded from the eastern part of the plateau to the hinterland under the low-temperature environment during the ice age. Therefore, different species have different responses to geoclimatic events, according to their genetic, physiological, and life-history characteristics. Amphibians, birds, and rodents are the most studied plateau animals. There are three main geographical distribution patterns in this area. (1) For two lineages, including *Pseudopodoces humilis* [9,10], or blood pheasant (*Ithaginis cruentus*) [11], and the alpine pygmy frog (*Nanorana parkeri*) [12], the differentiation was mainly caused by the high mountain barrier and the 400 mm precipitation boundary [12]. (2) The differentiation of four branches, including *Eospalax fontanierii* and the plateau pika (*Ochotona curzoniae*) [13,14], occurred mainly due to geographical barriers, such as mountains and basins. (3) Multiple differentiation clades, including *Phrynocephalus vlangalii* [15], showed no obvious differentiation patterns in the plateau region. Examples of these include the Blanford snow finch (*Pyrgilauda blanfordi*), the red-necked snow finch (*Pyrgilauda ruficollis*), the white-rumped blood finch (*Onychostruthus taczanowskii*), and the Tibetan antelope (*Pantholops hodgsonii*) [16–19].

The plateau pika is a small, non-hibernating mammal, also known as the black-lipped pika, that belongs to the family *Ochotonidae* [20], which is an important high-altitude model animal that is broadly distributed in the area 3000–6000 m above sea level [21]. Studies have shown that the survival footprint of the pika in China can be traced back to the early Eocene, about 53 Mya ago [22]. It was a relict species in the Quaternary glacial period and is an ideal organism for studying the relationship between historical events and biological evolution [23]. Most research on the lineage differentiation and distribution patterns of the pika has focused on the North American pika (*Ochotona princeps*) and the spotted necked pika (*Ochotona collaris*) [24–28].

In recent years, with the further development of pedigree geography, the differentiation of pikas on the Qinghai-Tibet Plateau has also received extensive attention. For

example, Yu et al. [29], who constructed a phylogenetic tree using mitochondrial genes (*Cytb* and *ND4*), found that pikas could be differentiated into three subgenera: Pika, Ochotona, and Conothoa. In the work of Niu et al. [30], the phylogenetic relationships of 27 species within the genus *Ochotona* were reconstructed based on the mitochondrial cytochrome b gene, with results showing that pikas are divided into five major species groups: the northern group, the surrounding Qinghai-Tibet Plateau group, the Qinghai-Tibet Plateau group, the Huanghe group, and the Central Asia group. Melo-Ferreira et al. [31] studied 12 nuclear genes from 11 representative species of pikas and concluded that there were four subgenera of pikas. Koju et al. [32] used two mitochondrial (*Cytb* and *COI*) and five nuclear gene segments (*RAG1*, *RAG2*, *TTN*, *OXAIL*, and *IL1RAPL1*) to infer the phylogeny of 14 taxa, and their results supported the *O. syrinx* group as a distinct lineage beyond the four recognized subgenera, leading the authors to conclude that there are five subgenera of pikas. In the same year, Liu et al. [33] conducted a comprehensive study of 27 existing species of pikas in China, combining morphological and molecular markers, and their results also supported the claim that the pikas are divided into five subgenera, namely, *Alienauroa*, *Conothoa*, *Ochotona*, *Lagotona*, and Pika. A study by Lissovsky [34] represents the most systematic molecular phylogenetic study of this topic. He consulted pika specimens from major museums in Europe and the United States, carried out molecular systematic studies, and combined these findings with morphological information, and believed that there were 28 species of pika in the world and performed consistent taxonomic revisions for all widely distributed species. In addition to the above studies on the intra-genus differentiation of pikas, a few scholars have also studied the intraspecific differentiation of Plateau pikas using molecular markers. For instance, Ci et al. [35] divided 32 plateau pika populations into 10 units according to geographical barriers such as mountains, rivers, and basins, and used mitochondrial genes to study the intra-specific differentiation of these 10 units. They found that the genetic diversity of these 10 units was high, and they were divided into six groups under the influence of climate and geographical barriers during the ice age. The central group was located in the hinterland of the plateau, with the five surrounding groups located around it. Then, He et al. [14] constructed a Bayesian phylogenetic tree using mitochondrial genes and divided 37 geographical populations of plateau pikas into four lineages, namely, east, west, north, and south. They found that the east, west, and north lineages had expanded in the past, whereas the southern lineages (located south of the Yarlung Zangbo River) had experienced a sharp decline. They speculated that this may have been related to the climate changes that occurred during the Quaternary glacial and interglacial periods.

Currently, there are many studies available on the differentiation of the pika family but few studies have focused on the evolution within the plateau pika population. Moreover, the existing intra-species research has been carried out using mitochondrial genes as molecular markers because mitochondrial genes are widely used in molecular phylogeography due to their unique maternal genetic characteristics. However, Avise et al. [36] suggested that since nuclear genes offer richer genetic information, they should be gradually adopted, and research that combines the use of multiple nuclear gene loci represents a promising direction for future phylogeographical research. Therefore, some scholars, such as Chen et al. [37], have studied the phylogeography of *Sigmella biguttata* using mitochondrial (*COI*, *COII*, and *ND1*) and nuclear genes (*ITS*). In addition, Jeremy et al. [38] studied the origins of family members in New Zealand using mitochondrial and nuclear genes. Moreover, Ding et al. [39] used mitochondrial (*COI*, *COIII*, *Cytb*, and *D-loop*) and nuclear genes (*vWF*) to study the intraspecific and intermediate differentiation of Tibetan hamsters. However, no studies hve been conducted on the differentiation of the plateau pika population using both nuclear and mitochondrial genes. To fill this gap, in this study, we used four mitochondria and three nuclear genes to investigate the intraspecies differentiation characteristics of the plateau pika. We aimed to determine the environmental factors that drove the plateau pika's evolution and diversity by understanding the genetic diversity, differentiation, and gene flow between the different populations; the findings of this study

will provide a scientific basis for the genetic management and resource conservation of the plateau pika population.

## 2. Materials and Methods

### 2.1. Sample Sources

A total of 302 samples (Table 1) were collected from 42 sample sites based on their geographical distances and geographical barriers (mountains, rivers, and basins) (Figure 1). Each individual was anesthetized on site after measuring its basic biological information, such as its body length and weight, and leg muscle tissue was removed with a sterile scalpel. The sample was placed in an Eppendorf tube and stored in liquid nitrogen.

**Table 1.** Sampling information and genetic parameters based on the combined dataset (*Cytb* + *COI* + *12S rRNA* + *D-loop* + *RAG1* + *IRBP* + *GHR*) of plateau pika geographic populations.

| Groups | Sample Point Code | Population | Longitude | Latitude | Altitude (m) | Sample Size | Number of Haplotypes $N_h$ | Haplotype Diversity $H_d$ | Nucleotide Diversity $\pi$ |
|---|---|---|---|---|---|---|---|---|---|
| Group 1 | LKZ | Langkazi | 90.417 | 29.109 | 4425 | 3 | 3 | 1.000 | 0.00518 |
| | JZ | Jiangzi | 90.101 | 28.901 | 4660 | 7 | 5 | 0.905 | 0.00077 |
| Group 2 | BDQ | Budongquan | 39.897 | 35.522 | 4610 | 6 | 6 | 1.000 | 0.00491 |
| | TTH | Tuotuohe | 92.44 | 34.216 | 4540 | 7 | 7 | 1.000 | 0.00407 |
| | NQU | Naqu | 91.797 | 31.28 | 4600 | 5 | 5 | 1.000 | 0.0028 |
| | AD | Anduo | 91.718 | 32.157 | 4799 | 7 | 7 | 1.000 | 0.00876 |
| | MZGK | Mozhugongka | 92.296 | 29.693 | 4434 | 12 | 12 | 1.000 | 0.00181 |
| | BLH | Beiluhe | 92.942 | 34.862 | 4590 | 5 | 5 | 1.000 | 0.00515 |
| | T1 | Tibet-1 | 87.218 | 29.237 | 4481 | 5 | 5 | 1.000 | 0.00167 |
| | T3 | Tibet-3 | 82.563 | 30.578 | 4944 | 3 | 2 | 0.667 | 0.00048 |
| | T4 | Tibet-4 | 85.089 | 29.493 | 4607 | 5 | 4 | 0.900 | 0.00055 |
| | NM | Nimo | 90.27 | 29.502 | 3908 | 6 | 6 | 1.000 | 0.00205 |
| | GLDD | Geladandong | 91.652 | 33.589 | 4873 | 4 | 4 | 1.000 | 0.0023 |
| | KEQK | Kaerqiuka | 90.755 | 37.043 | 4184 | 5 | 5 | 1.000 | 0.00477 |
| | AJKH | Ajikehu | 88.61 | 37.003 | 4250 | 5 | 5 | 1.000 | 0.01159 |
| | TZH | Tuzihu | 87.308 | 36.8 | 4750 | 5 | 5 | 1.000 | 0.00466 |
| Group 3 | MY | Menyuan | 101.275 | 37.69 | 3260 | 15 | 15 | 1.000 | 0.0062 |
| | ML | Mole | 100.299 | 37.963 | 3790 | 15 | 14 | 0.990 | 0.00235 |
| | TJ | Tianjun | 99.106 | 37.245 | 3370 | 11 | 11 | 1.000 | 0.00496 |
| | QLEB | Qilianebao | 100.934 | 37.968 | 3429 | 7 | 7 | 1.000 | 0.00223 |
| | QLAR | Qilianarou | 100.525 | 38.048 | 3031 | 2 | 2 | 1.000 | 0.00129 |
| | RS | Reshui | 100.434 | 37.548 | 3520 | 5 | 5 | 1.000 | 0.00276 |
| | GC | Gangcha | 100.134 | 37.325 | 3370 | 5 | 4 | 0.900 | 0.00216 |
| | ND | Niaodao | 99.758 | 37.171 | 3158 | 3 | 3 | 1.000 | 0.00556 |
| | JXG | Jiangxigou | 100.211 | 36.621 | 3157 | 5 | 5 | 1.000 | 0.02028 |
| | GD1 | Guide-1 | 102.067 | 37.2 | 3725 | 2 | 2 | 1.000 | 0.00575 |
| | GD2 | Guide-2 | 101.205 | 36.254 | 3650 | 6 | 6 | 1.000 | 0.00854 |
| Group 4 | NQ | Nangqian | 96.508 | 32.19 | 3620 | 12 | 12 | 1.000 | 0.00452 |
| | BS | Basu | 97.206 | 30.674 | 4490 | 5 | 5 | 1.000 | 0.00109 |
| | YLS | Yelashan | 97.295 | 30.187 | 4338 | 5 | 5 | 1.000 | 0.00104 |
| | BD | Bangda | 97.129 | 30.529 | 4348 | 4 | 4 | 1.000 | 0.00065 |
| Group 5 | TR | Tongren | 101.716 | 35.586 | 3813 | 5 | 5 | 1.000 | 0.00247 |
| | ZK | Zeku | 101.47 | 35.056 | 3690 | 12 | 12 | 1.000 | 0.0028 |
| | KA | Heka | 99.908 | 35.821 | 3890 | 5 | 5 | 1.000 | 0.00818 |
| | MQ | Maqin | 100.212 | 34.505 | 3720 | 12 | 12 | 1.000 | 0.00446 |
| | GanD | Gande | 100.218 | 34.203 | 4210 | 11 | 11 | 1.000 | 0.00295 |
| | AB | Aba | 101.581 | 33.009 | 3440 | 13 | 12 | 0.987 | 0.00113 |
| | SQ | Shiqu | 98.047 | 32.984 | 4400 | 10 | 10 | 1.000 | 0.00421 |
| | YS | Yushu | 96.886 | 33.057 | 3840 | 15 | 15 | 1.000 | 0.01348 |
| | ZD | Zhiduo | 95.696 | 33.939 | 4170 | 9 | 9 | 1.000 | 0.00305 |
| | QML | Qumalai | 95.877 | 34.139 | 4390 | 7 | 6 | 0.952 | 0.00784 |
| | MD | Maduo | 98.133 | 34.796 | 4250 | 11 | 11 | 1.000 | 0.00753 |

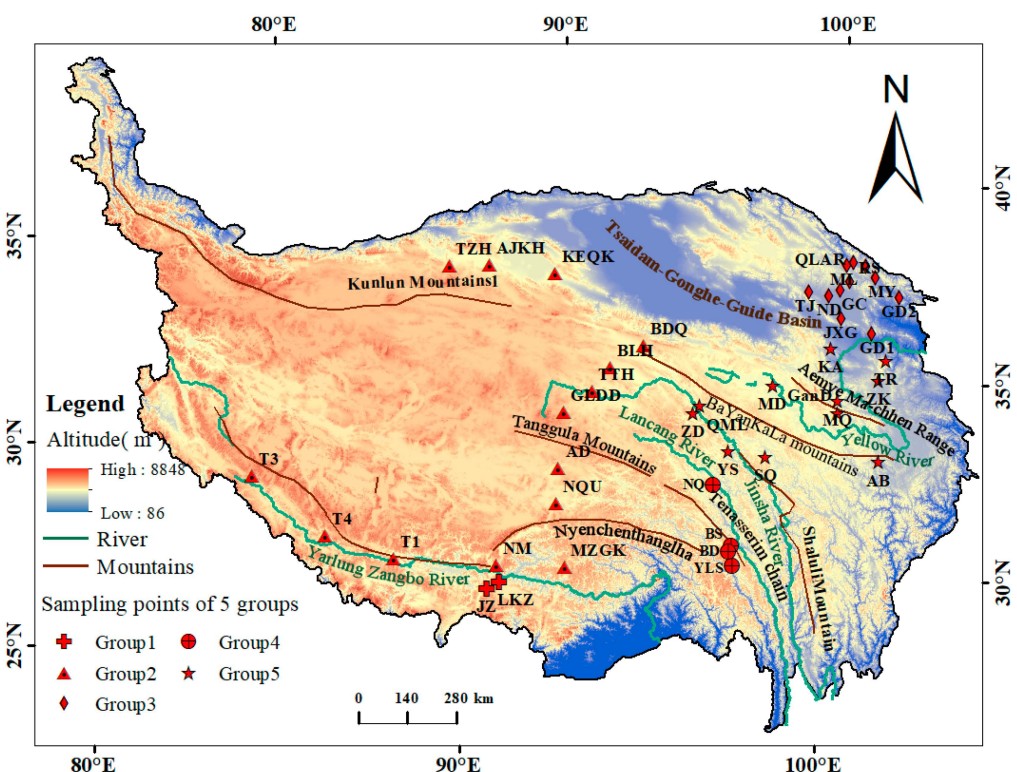

**Figure 1.** Map of sample locations for plateau pikas.

### 2.2. Genomic DNA Extraction, Polymerase Chain Reaction Amplification, and Sequencing

The genomic DNA was extracted from the plateau pika samples using a TaKaRa DNA Kit, and its concentration and integrity were detected using 2% agar-agar gel electrophoresis. The samples were stored at −80 °C. The primer design and sources are shown in Table 2. The polymerase chain reaction (PCR) volume was 50 μL, which contained 5.0 μL 10 × Taq buffer (Takara, Beijing China), 2.5 U Taq DNA polymerase, 1.5 mM MgCl$_2$, 0.1 mM deoxynucleotide triphosphates, 0.4 uM of each primer, and 0.5 μL total DNA. The rest of the volume was made up to 50 μL with sterilized ultra-purified water. The PCR conditions were as follows: predenaturation at 95 °C for 8 min; 35 cycles of denaturation at 95 °C for 45 s, annealing at 52 °C–64 °C for 45 s, and extension at 72 °C for 1 min; and a final extension at 72 °C for 7 min. The PCR products were photographed on a 1% agar gel to detect the target fragment, and they were sent to Sangon Biotech (Shanghai) Co., Ltd. for purification and sequencing.

### 2.3. Population Genetic Structure and Divergence

The nucleic acid sequences that were obtained via sequencing were spliced using DNAMAN 7 [14], and MEGA 7 [40] was used for sequence alignment and correction and to determine the base composition statistics. DnaSP 5.0 [41] was used to calculate the variable sites, parsimony-informative sites, the number of haplotypes ($N_h$), haplotype diversity ($H_d$), and nucleotide diversity ($\pi$).

SAMOVA 1.0 [42] was used for spatial analysis of the molecular variance, and the K value was set at 2–42. Each run was repeated 100 times. Then, an analysis of molecular variance (AMOVA) was conducted in Arlequin 3.5 [43] to estimate the distribution of the population's genetic distance and genetic variation and to calculate the population's genetic differentiation index ($F_{st}$). The maximum gene flow ($N_m$) was calculated using Formula (1) [37,44].

$$N_m = \frac{1 - F_{st}}{4F_{st}} \tag{1}$$

Govindajaru [45] divided gene flow into three levels according to the $N_m$ value: when $N_m > 1$, the gene flow among the various populations is considered to be high; when $0.25 < N_m < 0.99$, it is at a medium level; and when $N_m < 0.25$, it is low.

**Table 2.** Primers used in this study.

| Genes | Primer Sequences (5′-3′) | Amplified Size (bp) | Source | Annealing Temperature (°C) |
|---|---|---|---|---|
| *COI* | H:ACTACTGGCTTCAATCTACTTCTC<br>L:AAGACATAGAGGTTATGGAGTTGG | 1705 | Ding, 2020 [34] | 64 |
| *12S rRNA* | F:AAAGCAAAACACTGAAAATG<br>R:TTTCATCTTTTCCTTGCGGTA | 1149 | He, 2019 [15] | 64 |
| *Cytb* | H:CGGAATTCCATTTTTGGTTTACAAGAC<br>L:CGAAGCTTGATATGAAAAACCATCGTTG | 1170 | Kocher, 1989 [36] | 64 |
| *D-loop* | F:ATGTTCCGCCCAATCAGCCAAT<br>R:GTTGCTGGTTTCACGGAGGATGG | 655 | Ci, 2009 [31] | 59 |
| *GHR* | F:TCAGCCACAGAGGTTAGAAGG<br>R: CACATAGCCACACGATGAGAG | 665 | This study | 52 |
| *RAG1* | F:CCGACACCACCAACATTCAA<br>R:CCTTCACATCTCCACCTTCTTC | 1069 | This study | 52 |
| *IRBP* | H:GTCCTCTTGGATAACTACTGCTT<br>L:CTCCACTGCCCTCCCATGTCT | 744 | Ding, 2020 [34] | 61 |

Note: *12S rRNA*, 12S ribosomal RNA gene; *COI*, cytochrome oxidase subunit 1 gene; *Cytb*, cytochrome b gene; *D-loop*, control region; *IRBP*, interphotoreceptor retinoid-binding protein gene; *RAG1*, recombination activating gene 1 gene; *GHR*, growth hormone receptor gene.

### 2.4. Phylogenetic Analysis and Haplotype Network

The North American pika (*Ochotona princeps*) and European rabbit (*Oryctolagus cuniculus*) were used as the outgroups. The maximum likelihood (ML) and Bayesian inference (BI) methods were used to construct a phylogenetic tree with 302 tandem DNA sequences (*12S rRNA*, *COI*, *D-loop*, *Cytb*, *GHR*, *IRBP*, and *RAG1*). The maximum likelihood phylogenetic tree was implemented with MEGA 7. The best nucleotide replacement models (*12S rRNA*, *COI*, *D-loop*, *Cytb*, and *RAG1*: GTR + I + G; *GHR*: GTR + G; *IRBP*: HKY + I) were identified using the corrected Akaike information criterion [46], which was implemented in MrModeltest 3.7 [47]. Bayesian analysis was conducted using MrBayes 3.2.6, which was run with twenty million generations of Markov chain Monte Carlo (MCMC) simulation and sampled every 1000 generations. MCMC was run using the default model parameters, starting from a random tree. The first 25% were discarded as a conservative burn-in, and the remaining samples were used to generate a 50% majority-rule consensus tree. Here, a Bayesian posterior probability equal to or above 0.95 was considered to indicate strong relationships [48]. Subsequently, the tree file was visualized using FigTree 1.4.7 (http://tree.bio.ed.ac.uk/software/figtree/ accessed on 5 October 2022). Finally, the haplotype network diagram was constructed using a median-joining network by means of PopART 1.7 [49].

### 2.5. Nucleotide Mutation Rate and Estimation of the Divergence Time

To acquire as accurate a divergence time as possible, the number of nucleotide substitutions per site (d) was estimated basd on comparisons between the focal species and outgroup species using the following Formula (2):

$$d = (tv + tvR)/m \tag{2}$$

where tv is the number of transversions between *Ochotona curzoniae* and the outgroup taxa, R is the transition/transversion ratio within the *Ochotona curzoniae* complex, and m is the sequence length [50,51]. The transition and transversion values were calculated in the

program MEGA 7.0 [52]. The rates of nucleotide substitutions per site, per lineage, and per year were calculated using the formula $\lambda = d/2T$ when an estimate of d was obtained, where *T* is the divergence time between the ingroup and outgroup species [51]. The mutation rate per nucleotide site and per generation was calculated with the formula $\mu = \lambda g$, where g is the generation time (g = 0.46 years for *Ochotona curzoniae*) [53,54]. In this study, d was 0.0172 (tv = 17, R = 6.0, m = 6909) for the combined genes (*12S rRNA*, *COI*, *D-loop*, *Cytb*, *GHR*, *IRBP*, and *RAG1*). The rate of nucleotide substitution per site, per lineage, and per year ($\lambda$) was about $0.86 \times 10^{-8}$ (*T* = 10.65 Mya, obtained from Rui et al. [55]), and the mutation rate per generation ($\mu$) was about 0.396% $Mya^{-1}$ for the concatenated DNA sequence of *Ochotona curzoniae*.

The divergence times among the different populations of *Ochotona curzoniae* were estimated using BEAST 2.4.7 [56]. There were 6069 bp in the combined genes (*12S rRNA*, *COI*, *D-loop*, *Cytb*, *GHR*, *IRBP*, and *RAG1*). The best base replacement model was the GTR + I + G model, and the molecular clock model used was the strict clock model. The Markov chain Monte Carlo (MCMC) analysis was run twice, the chain lengths were 50 million, with sampling occurred every 1000 generations, and the first 10% was discarded as burn-in. The program TRACER 1.7 [57] was used to test the validity of the sampling results for MrBayes and BEAST (effective sample size [ESS] > 200). Finally, the tree files were visualized using iTOL (https://itol.embl.de/itol.cgi accessed on 15 October 2022).

*2.6. Demographic History*

Arlequin 3.5 [43] was used to calculate the mismatch distribution, Tajima's *D*, and Fu's *Fs* values to check whether the population had experienced historical expansion. The goodness of fit between the nucleotide mismatch distribution and the expected distribution under the population expansion model was detected using the sum of squared deviation (*SSD*) and Hapending's raggedness index (*Hri*). The Bayesian MCMC method in BEAST 2.4.7 [56] was used to speculate regarding the time of divergence for the different pedigrees, and the historical population dynamics were analyzed for the combined plateau pika DNA sequences using a Bayesian skyline plot (BSP). In this study, a strict clock was used in BEAST and the substitution rate was based on the estimated results of the nucleotide mutation rate, as described above. The same nucleotide substitution model was used for the Bayesian phylogenetic analysis. The BSP was run twice for MCMC chain lengths of 100 million generations, it was sampled every 1000 generations, and both the MCMC convergence and the ESS (ESS > 200) were tested using the program TRACER 1.7 [57]. Finally, R 3.6.3 was used to determine the composition.

**3. Results**

*3.1. Analysis of the Gene Sequence Variation*

For the seven gene sequences in this study, the variation analysis results are shown in Table 3. Among them, the *COI* gene exhibited the most total variation, with 150 variable loci, and the *IRBP* gene exhibited the least total variation, with 20 variable loci. The *D-loop* gene had the least single mutation sites (2), whereas the *GHR* gene had the most single mutation sites (26). In addition, the C + G content in the *IRBP* gene sequence was the highest (61.6%), and in the other genes, it was lower than the A + T content. The average conversion/transposition values for the seven genes were all greater than one. There were also only a few *IRBP* gene haplotypes, whereas the *D-loop* gene had the most haplotypes, with totals of 22 and 108, respectively. The concatenated sequence was 6909 bp in length and contained 773 mutation sites, 690 of which were parsimony-informative sites and 83 were single mutation sites. The base composition had a larger percentage of A + T (52.5%) than that of C + G (47.6%). Additionally, there were 290 different haplotypes, and the $\pi$ and $H_d$ values were 0.9997 and 0.01205, respectively.

**Table 3.** Analysis of different gene sequences and total sequence variation.

| Gene | *12S rRNA* | *Cytb* | *COI* | *D-loop* | *GHR* | *IRBP* | *RAG1* | Combined Sequence |
|---|---|---|---|---|---|---|---|---|
| Sequence length/bp | 1093 | 1158 | 1705 | 656 | 573 | 744 | 980 | 6909 |
| Variable sites | 114 | 150 | 277 | 143 | 40 | 20 | 29 | 773 |
| Singleton variable sites | 26 | 8 | 8 | 2 | 26 | 5 | 8 | 83 |
| Parsimony informative sites | 88 | 142 | 269 | 141 | 14 | 15 | 21 | 690 |
| Average conversion/transpose value | 2.6 | 10.7 | 5.5 | 6.3 | 2.6 | 9.4 | 5.4 | 6 |
| Haplotype | 73 | 80 | 92 | 108 | 28 | 22 | 72 | 290 |
| Haplotype/Sample number/% | 24.2 | 26.5 | 30.5 | 35.8 | 9.3 | 7.3 | 23.8 | 96 |
| A/% | 35.4 | 26.5 | 26.7 | 34 | 23.2 | 19.8 | 28.7 | 28.1 |
| T/% | 23 | 26.2 | 28.5 | 26 | 20.1 | 18.7 | 22.9 | 24.4 |
| C/% | 24.9 | 34 | 27.8 | 29.9 | 32.4 | 30.8 | 24.2 | 28.8 |
| G/% | 16.6 | 13.3 | 17 | 10.1 | 24.2 | 30.8 | 24.1 | 18.8 |

Nuclear genes: *GHR, IRBP, RAG1*. Mitochondrial genes: *12S rRNA, Cytb, COI, D-loop*.

*3.2. Population Genetic Structure Analysis*

For the different geographical populations (Table 1), the nucleotide diversity of eight populations (Guide-2, Heka, Qumalai, Maduo, Anduo, AJikehu, Jiangxhu, and Yushu) was significantly higher than that of the other populations ($\pi > 0.007$). The SAMOVA analysis based on the total sequences showed that $F_{ct}$ increased between K = 2 and 5 and decreased between K = 6 and 42. When K = 5, $F_{ct}$ reached its maximum value ($F_{ct} = 0.50$). Therefore, the best groups for plateau pika population differentiation were determined to be Group 1–Group 5 (Table 4) as the genetic diversity of these five lineages was relatively high ($H_d > 0.9$; $\pi > 0.004$).

**Table 4.** Genetic diversity of each group of plateau pika populations based on the combined dataset (*12S rRNA + COI + D-loop +Cytb + GHR + IRBP + RAG1*).

| Group | Populations Included | Number of Haplotypes $N_h$ | Haplotype Diversity $H_d$ | Nucleotide Diversity $\pi$ |
|---|---|---|---|---|
| Group 1 | LKZ JZ | 8 | 0.956 | 0.01027 |
| Group 2 | NM TZH AJKH KEQK T1 T3 T4 BLH MZGK AD NQU TTH BDQ GLDD | 78 | 0.999 | 0.00625 |
| Group 3 | MY ML GD1 TJ ND GC RS QLAR QLEB JXG GD2 | 68 | 0.999 | 0.00708 |
| Group 4 | BS BD YLS NQ | 26 | 1.000 | 0.00432 |
| Group 5 | MQ MD GanD TR ZK KA AB SQ YS ZD QML | 114 | 1.000 | 0.00699 |

For population information, see Table 1. The same below.

The results of the AMOVA (Table 5) showed that the genetic variation of the combined plateau pika sequences without grouping revealed significant differentiation among the populations (61.71% of the variation) when compared with the level within the populations (38.29%). The genetic variation among the groups (49.84%) was greater than that within the populations (16.66%) and among the populations (33.50%) when the haplotypes were divided into five groups.

Two key indices were used to gauge the degree of population differentiation, namely, $F_{st}$ and $N_m$. When $0 < F_{st} < 0.05$, the degree of differentiation among the populations was small; when $0.05 < F_{st} < 0.15$, it was moderate; when $0.15 < F_{st} < 0.25$, it was high; and when $F_{st} > 0.25$, it was extremely high [58]. As shown in Figure 2a, the pairwise $F_{st}$ among the lineages ranged from 0.554 to 0.798, which was greater than 0.25, and they all reached a significant level ($p < 0.05$). Furthermore, the pairwise $F_{st}$ values among lineage 1 (Group 1) were larger than those of the other groups (all greater than 0.7). When

compared with the other groups, the $F_{st}$ value between Group 4 and Group 5 (0.554) was the smallest. Furthermore, the range of $F_{st}$ values among the 42 geographical populations was 0.187–0.979 (Figure 2b and Appendix A, Figure A1), and the $F_{st}$ values among the geographical populations within the other lineages were all small except for those of lineage 1 (Group 1) and lineage 4 (Group 4). The statistical results for the $N_m$ values showed that the $N_m$ values among the five lineages ranged from 0.063 to 0.201 (Figure 2c), and this indicates that the maximum gene flow between the five lineages was low. The maximum gene flow within the geographical populations in lineages 2 (Group 2), 3 (Group 3), and 5 (Group 5) was larger, whereas the maximum gene flow within the geographical populations in lineages 1 (Group 1) and 4 (Group 4) was smaller than that in the other lineages (Figure 2d and Appendix A, Figure A2). Moreover, the pairwise $F_{st}$ values within and among eight populations (Ajikehu, Anduo, Jiangxigou, Guide-2, Qumalai, Yushu, Maduo, and Heka) and various populations were compared with various groups, and they were determined to be small. In addition, the $N_m$ values were all large (Figure 2d and Appendix A, Figure A2), indicating that there was a large amount of gene flow among these eight populations and the other lineages.

**Table 5.** AMOVA analysis results of the populations of plateau pika based on the combined dataset (*12S rRNA + COI + D-loop +Cytb + GHR + IRBP + RAG1*).

| | Source of Variation | D.f | Sum of Squares | Variance Components | Variation Percentage | Fixation Indices |
|---|---|---|---|---|---|---|
| One group | Among populations | 42 | 8578.158 | 26.71915 | 61.71 | $F_{st} = 0.61714$ |
| | Within populations | 261 | 4326.296 | 16.57585 | 38.29 | |
| | Total | 303 | 12,904.454 | 43.29499 | | |
| Five groups based on lineages | Among groups | 4 | 5752.325 | 24.74701va | 49.84 | $F_{sc} = 0.33220$ |
| | Among populations within groups | 37 | 2774.028 | 8.27352vb | 16.66 | $F_{st} = 0.66503$ |
| | Within populations | 260 | 4324.296 | 16.63191vc | 33.50 | $F_{ct} = 0.49840$ |

*3.3. Phylogenetic Tree and Haplotype Network Diagram Analysis Results*

The topological structures of the phylogenetic trees that were obtained by means of the ML and BI methods were the same. The 42 geographical populations of the plateau pika were clustered into five lineages (Figure 3a), which was consistent with the SAMOVA analysis results. Among these, lineage 1 diverged the earliest and was separated from the other lineages, whereas lineage 4 and lineage 5 were the last to diverge and had the smallest genetic distances between them. The phylogenetic tree analysis results were consistent with those of the haplotype network diagram (Figure 3b), where the haplotype in Group 4 was located in the center of the haplotype network diagram. Except for the geographical populations of Zhidao and Qumalai, which had three shared haplotypes, the other haplotypes were unique to each geographical population, indicating a high degree of differentiation among the geographical populations.

*3.4. Analysis of the Historical Population Dynamics*

3.4.1. Neutrality Test

The neutral test results (Table 6) showed that the Tajiam's *D* and Fu's *Fs* values of Group 2, Group 3, and Group 5 were all negative. Therefore, this indicates that they have experienced significant historical expansion. This is consistent with the results of the BSP curve (Figure 4b) and the single-peak diagram of the nucleotide mismatch distribution (Figure 4c). The BSP curve showed that the populations of Group 2, Group 3, and Group 5 began to expand about 0.06 Mya ago, and the expansion slowed down about 0.02 Mya. Group 3 expanded dramatically for the second time about 0.01 Mya ago. Then, the Tajima's D and Fu's Fs values for Group 1 were both greater than zero, the nucleotide mismatch distribution did not show an obvious single peak, and the BSP curve showed no evidence of expansion. Both the Tajima's D and Fu's Fs values for Group 4 were negative, but they were

not significant. The distribution of the nucleotide mismatch took the form of a multi-peak graph, and the BSP curve did not indicate an obvious expansion trend. Therefore, this indicates that the plateau pika of Group 1 and Group 4 did not undergo any obvious historical expansion.

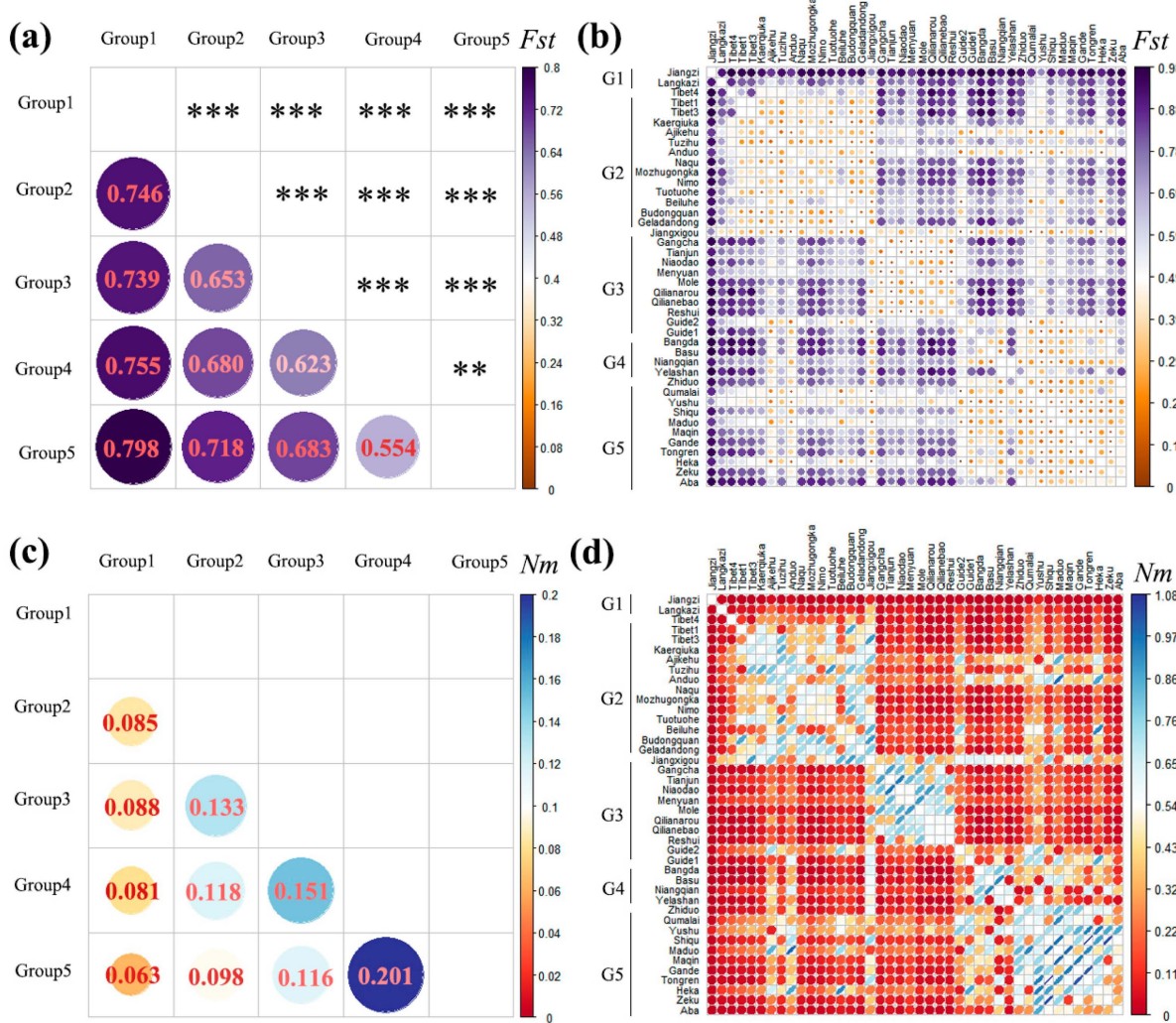

**Figure 2.** $F_{st}$ and $N_m$ values among populations and groups of plateau pika based on the combined dataset (*12S rRNA + COI + D-loop +Cytb + GHR + IRBP + RAG1*). (**a**) $F_{st}$ value of the group; the diagonal indicates the level of significance, ** $p < 0.01$; *** $p < 0.001$; (**b**) $F_{st}$ values of 42 populations, G1–G5: Group 1–Group 5; (**c**) $N_m$ value of group; (**d**) $N_m$ value of 42 populations, G1–G5: Group 1–Group 5.

**Table 6.** Neutral test and mismatch analysis of different lineages of plateau pika in the Qinghai-Tibet Plateau based on the combined dataset (*12S rRNA + COI + D-loop + Cytb + GHR + IRBP + RAG1*).

| Group | Tajima's D | Fu's Fs | SSD | Harpending's Raggedness Index r | T (Ma) |
|---|---|---|---|---|---|
| Group 1 | 0.98023 [ns] | 4.00851 [ns] | 0.89400 [ns] | 0.083333 [ns] | - |
| Group 2 | −1.14529 * | −24.19805 * | 0.84891 [ns] | 0.00086444 [ns] | 0.06 |
| Group 3 | −1.32911 * | −24.41101 * | 0.84603 [ns] | 0.00164939 [ns] | 0.06 |
| Group 4 | −0.83933 [ns] | −8.9991275 [ns] | 0.85944 [ns] | 0.01231735 [ns] | - |
| Group 5 | −1.65000 * | −24.16632 * | 0.88954 * | 0.00142879 [ns] | 0.06 |

SSD: Sum of squared deviations; [ns]: not significant; * $p < 0.05$; -: no increase.

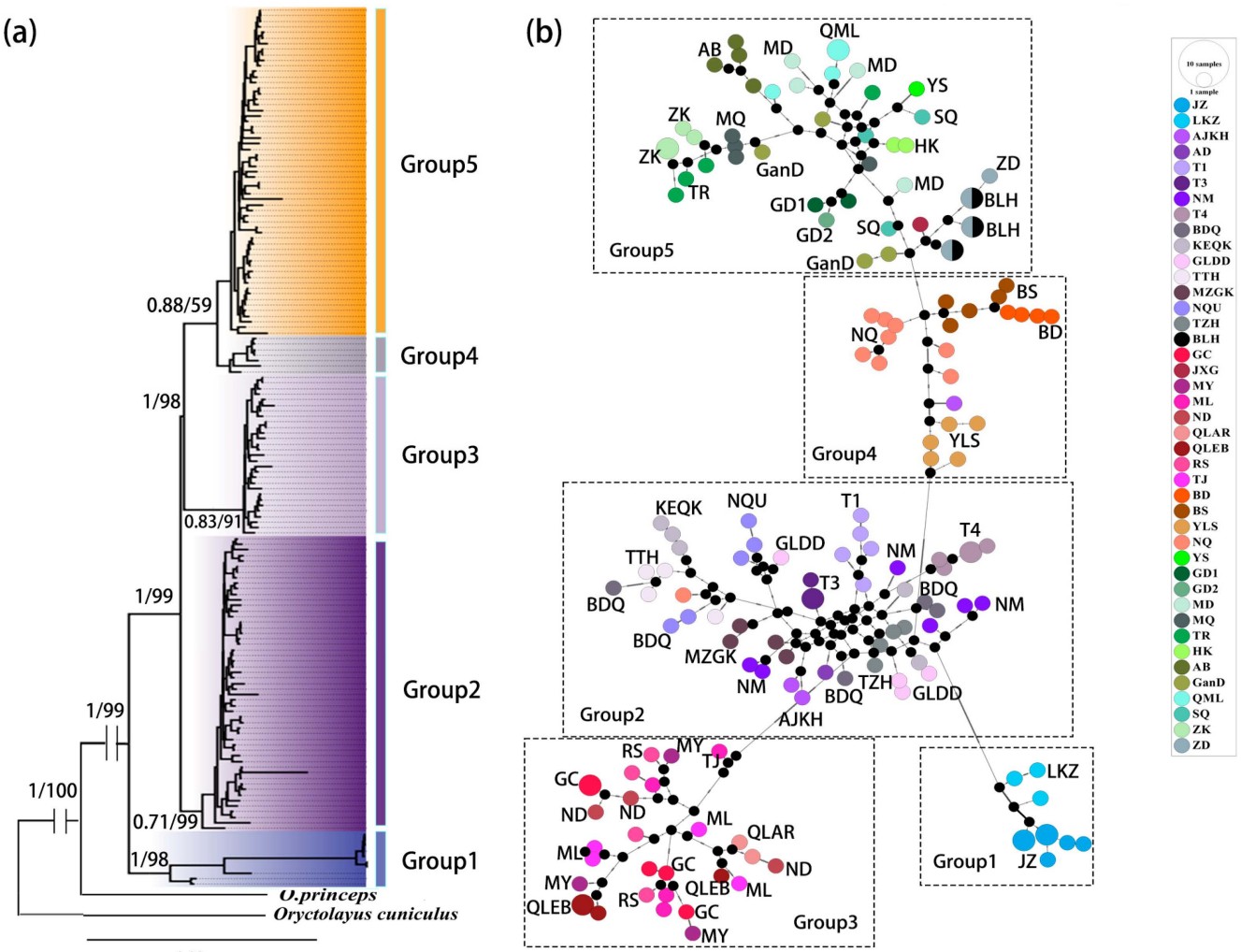

**Figure 3.** Maximum likelihood (ML) phylogenetic tree (**a**) and median-joining network (**b**) based on the combined dataset (*12S rRNA + COI + D-loop + Cytb + GHR + IRBP + RAG1*) of plateau pikas. Values near the node indicate the posterior probability values and bootstrap values of the phylogenetic tree. Short bars in the haplotype network represent mutation steps and the black dots represent missing haplotypes. The circle name and color correspond to different regions, and the haplotype circle size denotes the number of sampled individuals.

### 3.4.2. Estimation of the Historical Divergence Time

As shown in Figure 3a, the plateau pika population experienced four large differentiation events historically. Firstly, the populations south of the Yarlung Zangbo River (Group 1) underwent differentiation around 0.72 Mya (95% confidence interval HPD: 0.88–0.86 Mya). The second differentiation event occurred about 0.48 Mya (95% confidence interval HPD: 0.39–0.57 Mya), and the populations in the central Tibetan Plateau were differentiated (Group 2). The population in the northeastern Qinghai-Tibet plateau (Group 3) was estimated to have diverged 0.47 Mya (95% confidence interval HPD: 0.38–0.56 Mya). Additionally, the fourth differentiation occurred about 0.23 Mya (95% confidence interval HPD: 0.18–0.28 Mya) on both sides of the Lancang River (Group 4 and Group 5).

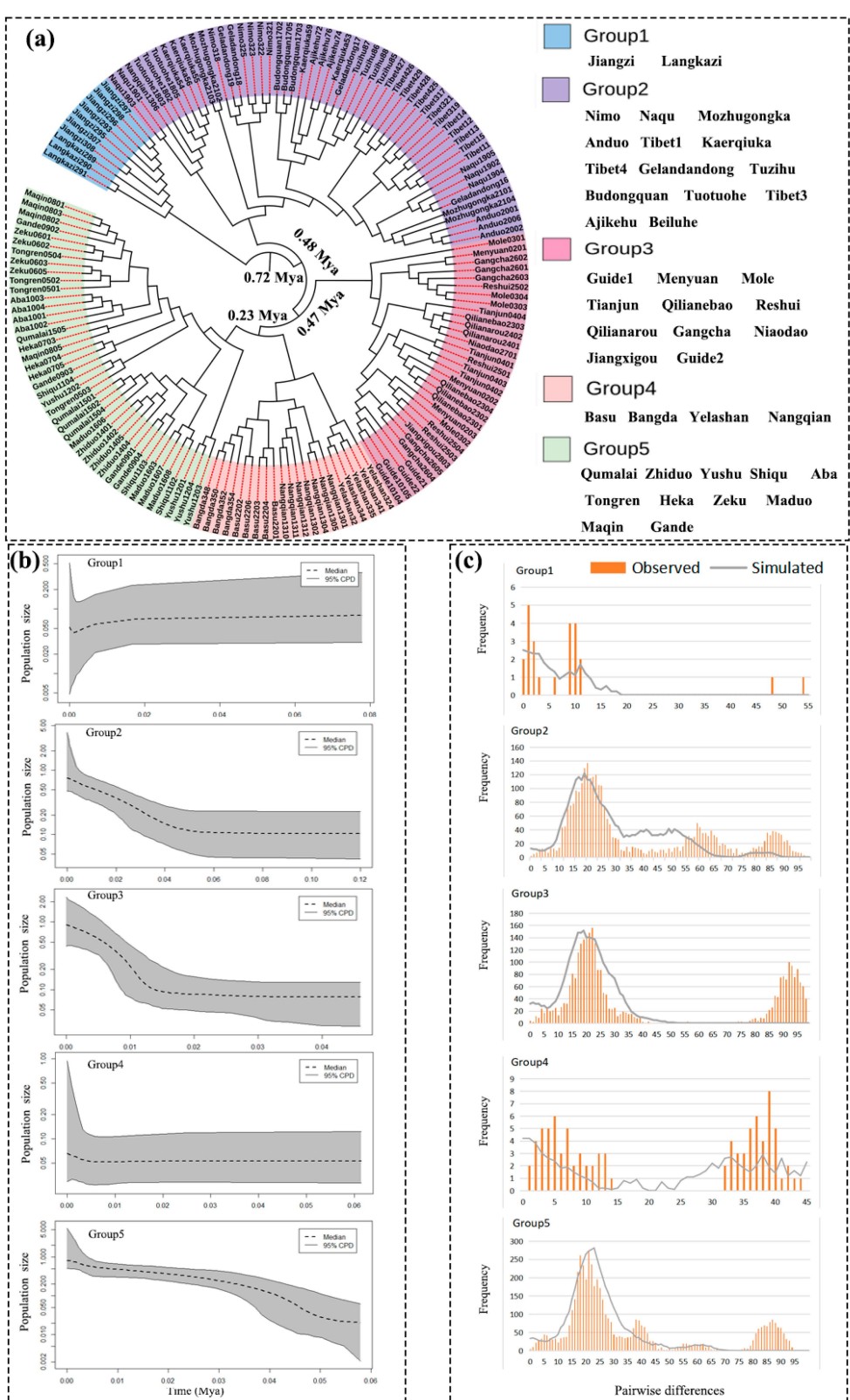

**Figure 4.** Historical dynamic analysis of plateau pika population. (**a**) Phylogenetic time-tree of plateau pika populations. (**b**) Bayesian skyline plots (BSPs) for population dynamics with the time of plateau pikas in the middle and lower reaches; dotted lines indicate median values and solid lines indicate 95% upper and lower confidence intervals. (**c**) Mismatch distribution analysis diagram of each group of plateau pika populations.

## 4. Discussion

*4.1. Effects of the Qinghai-Tibet Plateau Geographic Barrier on the Genetic Structure and Differentiation of the Plateau Pika Population*

Genetic diversity is an important indicator of the responses of biological populations to environmental change and human disturbance [59]. Intraspecific genetic diversity determines evolutionary capacity. Haplotype diversity and nucleotide diversity are important parameters that can be used to measure the genetic diversity of a population [60]. The results based on the combined sequences showed that the $H_d$ of the plateau pika was 0.9997, which was greater than 0.5, and $\pi$ was 0.01205, which was greater than 0.5%, indicating that the genetic diversity of the plateau pika was high on the Tibetan Plateau [61]. This is consistent with the research results of He [14], Ci [35], and others. On the one hand, the complex topography of the Qinghai-Tibet Plateau has fragmented the habitat of the plateau pika, and specific environmental factors in the different habitats could have driven the unique variation in its genetics and its evolution [62]. On the other hand, these findings might have been obtained because the plateau pika has a sizable number of geographical populations, with polygyny being its main form of reproduction [63], and dispersal behavior is common [64]. Dispersal improves the likelihood of mating across various families within a single geographic population, increasing genetic diversity. However, due to the relatively small dispersal range of the plateau pika, which basically remains within the range of 106–178 m$^2$ [65], the haplotype analysis results showed that there were only three shared haplotypes between Guide-2 and Tongren (Figure 2). The other 40 geographical populations did not share haplotypes, and each population had a high proportion of unique haplotypes. Thus, these diverse genetic resources are of positive significance to the future evolution of the plateau pika population.

The SAMOVA analysis and the phylogenetic tree showed that the 42 populations were divided into five lineages, which occupied different geographical areas (Figure 1). Lineage 1 (Group 1) was located south of the Brahmaputra River, Group 2 was in the hinterland of the Qinghai-Tibet Plateau, Group 3 was in the northeastern Tibetan Plateau, Group 4 was in the eastern part of the Tibetan Plateau, and Lineage 5 (Group 5) was in the Hengduan Mountain region. The AMOVA showed that the genetic variation was mainly identified within the lineages, at 49.84%, whereas the genetic variation within the populations was only 16.66%, and the genetic variation among the different geographical populations accounted for 33.5%. The $F_{st}$ and $N_m$ values among the different lineages were all greater than 0.25, and the $N_m$ values were all less than 0.25. The results showed that there was great genetic differentiation among the five lineages [66]. We hypothesized that the external cause of this variation and differentiation was mainly due to geographical isolation, which hindered gene exchange among the populations and resulted in the directional evolution of each lineage in each region.

As shown in Figure 1, the largest geographical barrier between Lineage 1 and the other lineages was the Yarlung Zangbo River, which was formed as early as the Late Miocene (about 8 Mya) [67] and which has one of the highest elevations for a river in China. The largest geographical barrier between Group 3 and the other lineages was the Qaidam-Gonghe-Guide Basin, which has an average elevation of 2600–9000 m and has very little vegetation coverage due to low precipitation and the soil texture. Therefore, it is not suitable for plateau pika survival, and it is the main geographical barrier that blocks gene exchange between Group 3 and the other lineages. The main geographical barrier between Group 4 and Group 5 was the Lancang River. The Lancang River was formed as early as 17 Mya [68], originating in Zadoi County, Qinghai Province. The valley has an elevation of 1000–4500 m and a width of 150–1200 m. It is a typical mountain-gorge landform [69]. Additionally, both the marshes in the river source area and the valleys in the middle and lower reaches are not suitable for plateau pikas, so they have become the biggest geographical barrier that blocks gene exchange between lineage 4 and lineage 5. The barrier effects of the Yarlung Zangbo River and the Qaidam-Gonghe-Guide Basin are reflected in the studies of He [14] and Ci [35] et al., and these may occur because the

pika is sensitive to low altitudes and high temperatures [27]. This is because low-altitude river valleys and basins have relatively high oxygen contents and temperatures, and this is inconsistent with the physiology of the plateau pika, which is adapted to survival at low altitudes and low temperatures and has developed over a long period [70]. Thus, these conditions result in a lack of gene exchange across the valleys and basins. Therefore, the barrier effect on plateau pika dispersal was prominent.

In contrast to the findings of studies by He and Ci et al. [14,35], the population differentiation of the plateau pika between the Hengduan Mountains (Group 4) and the eastern Qinghai-Tibet Plateau (Group 5) was obvious in this study, which may have occurred because those authors used mitochondrial genes to analyze population differentiation. In this study, both mitochondrial and nuclear genes were used as molecular markers to analyze the differentiation of the plateau pika population. Mitochondrial genes are maternal molecular markers, whereas nuclear genes are parental molecular markers, and plateau pika males are the main dispersers [62], so the results obtained when using both mitochondrial and nuclear genes were different from those obtained using mitochondrial genes alone. The $F_{st}$ and $N_m$ values (Figure 2) showed that there was a large amount of gene flow among the populations of lineages 1 (Group 1), 3 (Group 3), and 5 (Group 5), and the differentiation level was low. It is possible that the plateau pika underwent repeated dispersal, centered in a few refuges within each lineage in the Quaternary glacial and interglacial periods, leading to frequent gene exchange between the diverse populations within the lineage. The findings also indicated that geographical barriers such as the Tanggula, Nianqing Tanggula, and Kunlun Mountains, which are in the hinterland of the plateau, and the Bayan Har Mountains, Animaqing Snow Mountain, and Yellow River, which are in the eastern margin of the plateau, did not have significant effects on the isolation of the plateau pika. The gene flow among the four geographic populations within Group 4 was significantly smaller than that among the other three groups, probably because the populations of Group 4 were located in the Hengduan Mountain region, which has more mountains and valleys and more complex terrain. This could have restricted the dispersal of the plateau pika and resulted in less gene exchange among the geographic populations within Group 4. If this isolation continues to play a role, more differentiation will likely occur. This result is consistent with the fact that the Hengduan Mountain region, because of its favorable geographical and climatic conditions, led to rapid biological evolution and high biodiversity.

### 4.2. Phylogeny and Divergence Time of the Plateau Pika

In a haplotype network diagram, the original haplotype is usually positioned in the center of the stele structure [71]. In Figure 2b, the oldest lineage in the Qinghai-Tibet Plateau was Group 4, which was located in the Hengduan Mountains, a hotspot for biodiversity research. This area is also the place of origin of many species [72]. The Hengduan Mountains may have been the place of origin of the plateau pika, with the species then spreading over the whole plateau, gradually forming the existing distribution pattern. In the early stage of its formation (about 22 Mya), the Yarlung Zangbo River was just a tributary of the Red River [73]. The riverbed was shallow and did not fully form until about 8 Mya, and it experienced several periods of low flow due to drought. Therefore, the plateau pika may have undergone trans-river dispersal through some of the shallow areas during the low-flow period of the Yarlung Zangbo River [74]. Later, with the rise of the Tibetan Plateau, the Yarlung Zangbo River system continuously expanded to form a large canyon [2], blocking the gene flow of the plateau pika. Consequently, the lineage south of the Yarlung Zangbo River (Group 1) first differentiated about 0.72 Mya, which is a similar situation to that implied by the results of He et al. [14]. After several uplifting movements, the plateau's hinterland rose to 3000–3500 m about 0.62 Mya [2]. The eastern region of the plateau has a relatively low elevation due to the small impact of the uplifting movement. Then, lineage 2 (Group 2) differentiated around 0.48 Mya. Concurrently, the drought intensified in the northwest region of the plateau, and the Qaidam Basin and Gonghe Basin were successively formed, which blocked the gene exchange of the plateau pikas

that were distributed on both sides of the basin, resulting in the differentiation of lineage 3 (Group 3) around 0.47 Mya in the northeast region of the Plateau. In the Quaternary (about 2.48 Mya), the plateau uplifted further and experienced repeated cycles of glaciation and interglaciation [2]. These great changes in geology and climate further expanded the major water systems of the Jinsha River, Nujiang River, and Lancang River in the Hengduan Mountain region [68]. In addition, the further uplift of the major mountains, such as Thanyuntaweng, Mangkang, and Nianqing Tanggula, led to the continuous fragmentation of the plateau pika habitat in the Hengduan Mountains, and, finally, Group 4 and Group 5 were divided into two lineages about 0.23 Mya.

### 4.3. Historical Dynamics of the Plateau Pika Population

Tajima's D value, Fu's Fs value, the nucleotide mismatch distribution, SSD, Hri test results, and BSP diagrams are widely used in the analysis of population history dynamics. In general, when the distribution of the nucleotide mismatch presents a smooth single peak, the SSD and Hri tests do not deviate significantly from the population expansion model and the Tajima's D and Fu's Fs values are significantly negative. This is considered evidence of a historical population expansion [75–79]. In this study, the neutral testing and BSP analysis showed that Group 2, Group 3, and Group 5 expanded dramatically about 0.06 Mya, whereas Group 1 and Group 4 did not show similar fluctuations. This period was the interglacial period before the last glacial maximum (about 0.06–0.03 Mya) [2,80], where the temperature in the Tibetan Plateau rose, glaciers melted, the amount of vegetation increased, and the suitable area for plateau organisms increased. Consequently, during this time, the Group 2, Group 3, and Group 5 populations all experienced tremendous growth. In addition, the temperature of the Tibetan Plateau during this period was 1 °C higher than that of the Holocene Megathermal Period according to Yafeng et al. [2]. However, due to the low latitude and the annual average temperature in the southern region of the Qinghai-Tibet Plateau, which can be as high as 4–5 °C [81], the temperature in the southern region of the plateau may have reached 5–6 °C. However, the optimal annual average temperature for the plateau pika is −1–0 °C [82], and the species is sensitive to high temperatures [83,84]. Thus, Group 1 and Group 4, which are located in the southern region of the Qinghai-Tibet plateau, may have had a reduced fitness index due to the high temperatures at this time, so they did not undergo expansion like the other lineages. This result was similar to that of Ci et al. [35].

Yujiao et al. [85] showed that other plateau pika lineages, except for those south of the Yarlung Zangbo River, expanded rapidly between 0.076–0.017 Mya; however, after 0.017 Mya, the plateau pika population rapidly declined. During this period, the Qinghai-Tibet Plateau entered the peak of the ice age [2]. The temperature was 9 °C lower than that of modern times, the precipitation was only 30–70% of that in modern times, the climate deteriorated, the glacier extent increased, and the vegetation biomass decreased, restricting the expansion of the plateau pika. The results of this study showed that during this period (about 0.02 Mya), the plateau pika did not exhibit a downward trend but the expansion trend slowed down significantly or even stopped temporarily (Figure 4b), indicating that the climate change during the ice age impacted the survival of the plateau pika. At this time, the numbers of most of the plateau species declined, and the organisms in the hinterland of the plateau retreated to the refuges in the east of the plateau [6] but the cold-tolerant and drought-tolerant organisms, such as the golden dew plum and red sand, began to expand during the ice age [86–88].

The plateau pika is also a hardy creature, and it may have used its special physiology [70] to resist the cold, which is why it did not decline significantly during the glacial period. This may also have occurred because the populations of each lineage had their own refuges, such as Amdo and Ajikehu in the region where Group 2 is located. Jinguangou and Gude-2 in the region of Group 3 and Yushu, Heka, Maduo, and Qumalai in the region of Group 5 may also have been refuges for each lineage due to their high haplotypes and higher genetic diversity than those for the other regions for the same lineages [89,90]. Similar

findings were also found in the genealogies of other plants and animals. For example, Chen et al. [91] studied the genetic evolution of *Rhodiola* and found that the northern and southern Tanggula Mountains and Hengduan Mountains may be refuges. The Amdo area in this study is located in the southern region of the Tanggula Mountains. *Hippophae tibetana* [92] also has several refuges in the western, central, and southeastern parts of the plateau; *Potentilla glabra* [8] has refuges in the upper reaches of Lancang River and Hengduan Mountains. Moreover, the plateau zokor [13] also has four refuges near the origin of the Lancang River, the northeastern area of the Qinghai-Tibet Plateau, and the eastern part of the Qinghai-Tibet Plateau.

The results of the molecular variance analysis in this study also supported the notion that the existing distribution pattern of the plateau pika was formed due to its dispersal into multiple refuges. If the plateau pika retreated to the eastern part of the Qinghai-Tibet Plateau or the Hengduan Mountains during the ice age and gradually spread from the east to the hinterland of the plateau after the ice age (like other organisms) [93], the populations that are located further away from the eastern plateau and Hengduan Mountains would have decreased genetic diversity due to the long-distance dispersal [94] and the gene flow among the populations, and other geographic populations would be small. However, the molecular variance results in this study showed that the gene flow among the geographical populations within each lineage was large, and the gene flow among the different lineages was small. In particular, lineages 1, 2, and 3 had a higher differentiation level and smaller gene flow than that between lineages 4 and 5, which are located in the eastern part of the plateau and the Hengduan Mountain Range, indicating that there was still isolation among these lineages during the Quaternary glacial and interglacial cycles. Therefore, we speculate that the plateau pika first spread to the entire Qinghai-Tibet Plateau from the Hengduan Mountain Range. Then, small-scale back-migration and dispersal were carried out to multiple refuges in various regions of the plateau in response to climatic cycles during the glacial and interglacial periods.

## 5. Conclusions

In this study, we found that the genetic diversity of the plateau pika was generally high on the Tibetan Plateau ($H_d$ = 0.9997; $\pi$ = 0.01205), and the plateau pika evolved into five lineages that occupied different geographical areas, namely, lineage 1 (Group 1) in the south of the Yarlung Zangbo River, lineage 2 (Group 2) in the hinterland of the plateau, lineage 3 (Group 3) in the northeastern part of the plateau, lineage 4 (Group 4) in the Hengduan Mountains, and lineage 5 (Group 5) in the eastern part of the plateau. Our results also proved that the geographical barriers between the five lineages, such as the Yarlung Zangbo River, the Qaidam-Ghuong-Guide Basin, and the Lancang River, effectively promoted the population differentiation of the plateau pika. Other geographical barriers, such as the Jinsha River and Yellow River; Tanggula, Tanggula Nianqing, Kunlun, and Bayan Har Mountains; and Animaqing Snow Mountain, were less effective. We also found that the plateau pika first spread from the Hengduan Mountains to the entire Qinghai-Tibet Plateau and then carried out small-scale migration and dispersal into multiple refuges in various regions of the plateau in response to climate changes during the glacial and interglacial periods.

Except for Group 1 and Group 4, all the other populations underwent a rapid expansion between 0.06 and 0.01 Mya but their expansion was considerably delayed or halted by the effects of climate change during the last glacial maximum (0.02 Mya). In conclusion, the plateau pika in the Qinghai-Tibet Plateau exhibits high genetic diversity, and topographical obstacles including mountains, valleys, and basins created by the uplift of the plateau and climatic changes since the Quaternary have played an important role in the differentiation and historical dynamics of the plateau pika population.

**Author Contributions:** Y.Q. conducted the research, analyzed the data, and wrote the paper; X.P. made suggestions to this paper; Z.L. and D.S. helped to process the data; Z.C. guided the research and performed extensive updating of the manuscript. All authors have read and agreed to the published version of the manuscript.

**Funding:** This research study was funded by Qinghai Provincial Key Laboratory of Medicinal Animal and Plant Resources of the Qinghai-Tibetan Plateau.

**Institutional Review Board Statement:** The animals involved in the study were handled in accordance with the National Regulations on the Administration of Laboratory Animals (GB14923-2010).

**Data Availability Statement:** All data and materials are available upon request.

**Acknowledgments:** We are particularly indebted to Su Xu from Qinghai Normal University and Delin Qi from Qinghai University for their constructive suggestions regarding an earlier draft of this paper.

**Conflicts of Interest:** The authors declare no conflict of interest.

## Appendix A

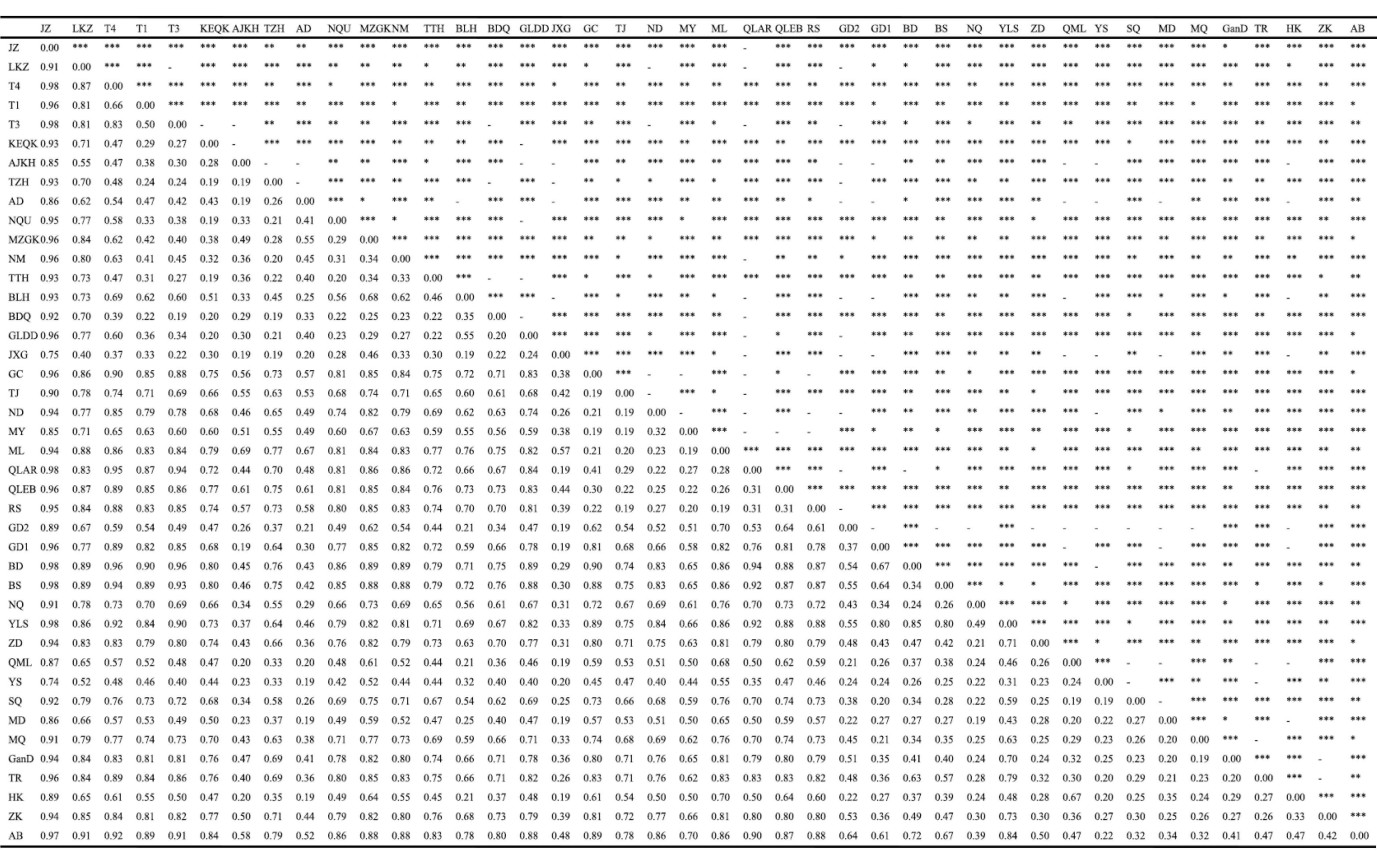

**Figure A1.** The fixation index ($F_{ST}$) results for 42 populations. The diagonal indicates the level of significance, * $p < 0.05$; ** $p < 0.01$; *** $p < 0.001$; -: not significant.

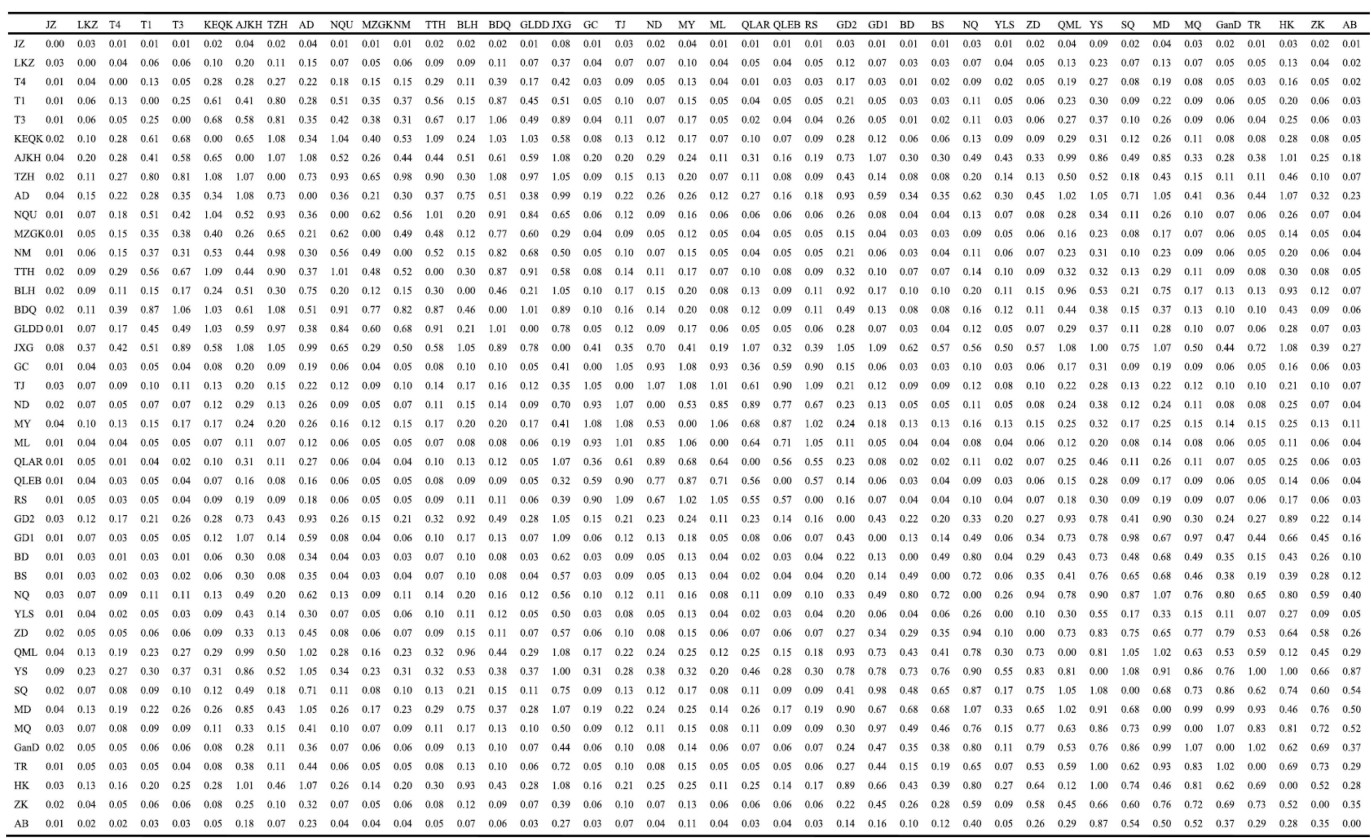

**Figure A2.** The maximum gene flow ($N_m$) results for 42 populations.

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
