# Peer review of "Phylogeography of the Plateau Pika (Ochotona curzoniae) in Response to the Uplift of the Qinghai-Tibet Plateau"

_diversity, doi:10.3390/d15020307_

Round 1
Reviewer 1 Report
Qi et al. explored the phylogeography of Ochotona curzoniae in Qinghai-Tibet Plateau using four mitochondrial and three nuclear genes. The study is interesting to the readers of Diversity. However, the MS requires careful revision and improvement.
Abstract has many colloquial words that have lowered its values. Consider revising.
The introduction section has a major focus on climatic cycles and the upliftment of the plateau. It could be shortened and emphasis could be given to the effects of such oscillations on animals living there. It should end with sound research questions/objectives.
The methods section, especially in phylogenetic tree and haplotype network analysis, requires extensive improvement.
Effects of different sample sizes from different populations in haplotype diversity, nucleotide diversity, # variable sites, etc. should be tested statistically.
Almost all the illustrations are not readable, please improve.
I have made many other comments in the annotated PDF.
All the best!

Author Response
Dear Professor
Thank you for your valuable suggestions. We have revised the paper in detail according to your suggestion.Please refer to the attachment for details.

Reviewer 2 Report
I read the paper " Phylogeography of the plateau pika(Ochotona curzoniae) in response to the uplift of the Qinghai-Tibet Plateau " by Yinglian et al. From my point of view, the paper is good, very interesting, and important in research of O. curzoniae. although it contains some flees, necessary to correct. The manuscript is acceptable for publication.
The manuscript needs improvement in the introduction and discussion. Lines 110-112 of the manuscript may mislead the research on pikas as there are several studies on the phylogeny and distribution of pika species except for the American and the collared pika. Therefore, the statement underestimates the research in pikas of Asia. Authors are suggested to review
Ø Feng, T., Kao, Y., 1974. Taxonomic notes on the Tibetan pika and allied species— including a new subspecies. Acta Zoologica Sinica 20, 76–87. Feng, Z., Cai, G., Zheng, C., 1986. The Mammals of Xinzang. Science
Ø Niu, Y., Wei, F., Li, M., Liu, X., Feng, Z., 2004. Phylogeny of pikas (Lagomorpha, Ochotona) inferred from mitochondrial cytochrome b sequences. Folia Zoologica-Praha 53, 141–156
Ø Yu, N., Zheng, C., Zhang, Y.-P., Li, W.-H., 2000. Molecular systematics of pikas (Genus Ochotona) inferred from mitochondrial DNA sequences. Mol. Phylogenet. Evol 16, 85–95
Ø Koju, N.P., He, K., Chalise, M.K., Ray, C., Chen, Z., Zhang, B., Wan, T., Chen, S. and Jiang, X., 2017. Multilocus approaches reveal underestimated species diversity and inter-specific gene flow in pikas (Ochotona) from southwestern China. Molecular phylogenetics and evolution, 107, pp.239-245.
Ø Lissovsky, A.A., Yatsentyuk, S.P. and Koju, N.P., 2019. Multilocus phylogeny and taxonomy of pikas of the subgenus Ochotona (Lagomorpha, Ochotonidae). Zoologica Scripta, 48(1), pp.1-16.
Ø Lissovsky, A., 2014. Taxonomic revision of pikas Ochotona (Lagomorpha, Mammalia) at the species level. Mammalia 78, 199–216.
Ø Lissovsky, A.A., Ivanova, N.V., Borisenko, A.V., 2007. Molecular phylogenetics and taxonomy of the subgenus Pika (Ochotona, Lagomorpha). J. Mammal. 88, 1195– 1204.
Ø Lissovsky, A.A., Yatsentyuk, S.P., Obolenskaya, E.V., Koju, N.P. and Ge, D., 2022. Diversification in highlands: phylogeny and taxonomy of pikas of the subgenus Conothoa (Lagomorpha, Ochotonidae). Zoologica Scripta, 51(3), pp.267-287.
Ø Liu, S., Jin, W., Liao, R., Sun, Z., Zeng, T., Fu, J., Liu, Y., Wang, X., Li, P., Tang, M., Chen, L., Dong, L., Han, M., & Gou, D. (2017). Phylogenetic study of Ochotona based on mitochondrial Cytb and morphology with a description of one new subgenus and five new species. Acta Theriologica Sinica, 37(1), 1–43. https://
doi.org/10.16829/j.slxb.201701001
Ø Ge, D., Wen, Z., Xia, L., Zhang, Z., Erbajeva, M., Huang, C. and Yang, Q., 2013. Evolutionary history of lagomorphs in response to global environmental change. PLoS One, 8(4), p.e59668.
Similarly, line 115-119 creates confusion, since the first reporting of five subgenera in pikas was done by Koju et al, (2017), and gave the name “syrinx complex” later in the same year by Liu et al (2017) also supported the existence of five genera in pikas with their respective name. In addition, Koju et al, (2017) reported mtDNA introgression between O. census to O. curzoniae and suggested taking caution to express phylogeny based on mtDNA.
In the result sections the figures are difficult to read especially figure 2 (b and d) and figure 3 b.
The plateau pika evolved into five lineages that occupied different geographical areas is the major result of the study and is particularly interesting. It would be nice if the authors could explore this further using a more targeted approach for the divergence of these possible groups i.e. derive important parameters such as effective population sizes and more precise estimates of the time of divergence. But still, the manuscript explains better geographical barriers between the five lineages and their divergence.
Nevertheless, I think this paper will be an important and interesting study after the corrections, mentioned above
Author Response

(The authors gave the same response as above.)

Reviewer 3 Report
In this manuscript, Qi and collaborators report the phylogeographic context for the origin and diversification of specific lineages of the plateau pika (Ochotona curzoniae) in the Qinghai-Tibet Plateau. Although the study is interesting and is not fatally flawed, I think it needs important revision before it can be accepted for publication.
My major concerns can be summarised in the two following points:
(1) In general, I do not consider that the quality of the presentation of the results is good enough. Pretty all figures of the manuscript present a low-quality resolution and definition that often makes difficult to see the different elements that appear in the figures. E.g.: In Figure 1 the name of the rivers cannot read well, there are problems with the colour based codes of the localities (please see my comments below), or in Figure 2 (specially Figure 2b and Figure 2d) it is impossible to read the Fst and Nm values for each of the 42 localities. Please see as well all my comments below about figures. Moreover, the figure captions are in general scarce in the description of the figure contents. Can the authors give some more details please?
(2) The authors assigned each of the 42 sampled localities to 5 different groups or lineages. What are they based on to do this assignment? Can they specify it explicitly in the main text of the manuscript? It is not clear to me from the text. As I understand, it appears that the authors use the results of the phylogenetic trees (ML, BI or both? Not specified) to describe these 5 groups. However, have in mind that from the phylogenetic trees, the only lineages that are clearly recovered and with very high bootstrap support are Groups 1 and 2. In the case of Group 3, the clade is recovered without support. Can the authors give the bootstrap value for this group to have an idea about the possible credibility of this group? I mean, if the BS value is between 50 and 70 (or over) is moderately supported and the group can be considered. However, if the BS value is below 50, I would not give much credibility to this group, specially in the results/discussion sections. On the other hand, Group 4 is recovered as paraphyletic to Group 5. Moreover, the trees suggested that there could be two different groups 4 that perhaps might coincide with my comment below about the distribution of Group 4 localities at both sides of the Lancang River? The results of the phylogenetic trees (recovered clades and groups) are very important and in some cases (specially for Groups 4 and 5) they are not matching the results of the rest of analyses i.e.: time-tree, gene flow, SAMOVAS, etc., which apparently suggest that groups 4 and 5 are real. Why do not the authors say nothing about this problematic? Why do the authors consider groups 3, 4 and 5 as valid? How can all this affect to the discussion of the phylogeographic events that have shaped the plateau pika distribution? Please, amend all this in the text.
Some other issues that the authors should address:
- When the authors refer to a specific gene, they should always use the same name for that specific gene i.e. the authors should use COI or COX1 but not a mixture of both names depending on the situation. Moreover, gene names should be always italicised. In case of doubts about which name to retain, a good guide to gene nomenclature can be found at HGNC (HUGO Gene Nomenclature Committe) https://www.genenames.org/
- Line 39: Perhaps 2.5 x 106 km2 could be changed to 2.5 million km2.
- Lines 105-106: Please change “family Ochotonidae, is an important high-altitude model animal and broadly distributed” to “family Ochotonidae AND is an important high-altitude model animal THAT IS broadly distributed”.
- Line 121: Please change “distributed on the Tibetan Plateu, found that the different” to “distributed on the Tibetan Plateu AND THEY found that the different”.
- Line 157: Is there any criteria for sampling localities selection? Can the authors specify a little bit more about their reasons for choosing the 42 localities that appear in the paper instead of other different localities?
- Figure 1: The colour based codes used to identify the 5 different groups or lineages of localities cannot be well distinguished. Please, chose other different type of codes (for example, different symbols better than same symbol with different colours). Where do the 5 different groups come from? This is applicable for Table 1 as well.
- Figure 2: In Figure 2b and 2d, values of Fst and Nm for populations cannot be read as occur in Figure 2a and 2c. Can the authors give these values in a Supplementary Table for example? Otherwise this information cannot be found anywhere.
- Figure 3: Which phylogenetic tree is represented? The ML tree or the BI tree? Please indicate. In the haplotype network, what do black dots indicate? Please specify. Moreover, the authors use colour based codes to illustrate the haplotypes from each of the different sampled localities (especially different intensities of a same colour for those localities of the same group/lineage) but this is very difficult to see. If the authors still want to show this information in the figure, they have to use other different type of codes (different combinations of symbols plus colours, the sample point codes as in Figure 1, etc). Also, the caption indicating the different colour based codes and localities do not read well. Finally, in line 330, authors should change “bootstep” to “bootstRAP”.
- Table 1: Why do not the authors add a new column to the table that showa to which of the 5 different group each of the 42 localities correspond to? This would really facilitate comparisons between Table 1 and Figure 1.
- Table 3: Perhaps indicate which are the mitochondrial genes and the nuclear ones as well? Please change “Parimony informative sites” to “ParSimony informative sites”. I recommend authors to specify that Haplotype refers to the total number of different haplotypes that have been found for each gene. Perhaps, as an extra variability information, the authors could add for each of the genes what percentage represent the total number of haplotypes respect to the total number of sampled individuals. E.g.: for Cytb there are 80 different haplotypes out of 302 total samples what represents the 26.5%.
- Table 4: It is a bit confused to identify which of the populations are included in each of the 5 different groups.
- Lines 279-284: Where do the values of the genetic variation among groups come from (i.e.: the 46.57%, 33.35% and 20.08%)? These values do not coincide with variation percentage values for groups of Table5.
- Lines 300-308: Where do the results come from? I cannot find the information anywhere.
- Line 310: Please change “Fst and Nm value among population and groups” to “Fst and Nm valueS among populationS and groups”.
- Line 357: Should not it be Figure 4a instead of Figure 3a?
- Lines 412-458: Why do authors sometimes use lineages and other times use groups? Why do not they use always the same nomenclature? Moreover, where is the Yarlung Zangbo River indicated in Figure 1? I could not find it. Actually, and according to Fig 1, it seems that is the Brahmaputra River the one that isolate localities from Group 1 to the others. Moreover, and according to Fig.1, it is not clear to me than the Lancang River was the main geographic barrier between Groups 4 and 5 since there are populations from Group 4 at both sides of that River (NQ, BS, BD and YLS at one side, while TR, ZK, MQ, and MD along with the localities of Group 5 are located at the other side of the Lancang River).
Author Response

(The authors gave the same response as above.)

Round 2
Reviewer 1 Report
Thank you for incorporating the suggestions.
All the best!